# Relationship between the Dietary Inflammatory Index Score and Cytokine Levels in Chinese Pregnant Women during the Second and Third Trimesters

**DOI:** 10.3390/nu15010194

**Published:** 2022-12-30

**Authors:** Tingkai Cui, Jingchao Zhang, Liyuan Liu, Wenjuan Xiong, Yuanyuan Su, Yu Han, Lei Gao, Zhiyi Qu, Xin Zhang

**Affiliations:** 1Department of Maternal, Child and Adolescence Health, School of Public Health, Tianjin Medical University, Tianjin 300070, China; 2Tianjin Key Laboratory of Environment, Nutrition, and Public Health, Tianjin Medical University, Tianjin 300070, China

**Keywords:** dietary inflammatory index, pregnant women, cytokines, food parameters

## Abstract

The impact of dietary inflammatory potential on serum cytokine concentrations in second and third trimesters of Chinese pregnant women is not clear. A total of 175 pregnant women from the Tianjin Maternal and Child Health Education and Service Cohort (TMCHESC) were included. The dietary inflammatory index (DII) was calculated based on 24-h food records. Serum tumor necrosis factor-α (TNF-α), interleukin 1β (IL-1β), IL-6, IL-8, IL-10, C-reactive protein (CRP), and monocyte chemoattractant protein-1 (MCP-1) levels in the second and third trimesters were measured. The mean DII scores (mean ± SD) were −0.07 ± 1.65 and 0.06 ± 1.65 in the second and third trimesters, respectively. In the third trimester, IL-1β (*p* = 0.039) and MCP-1 (*p* = 0.035) levels decreased and then increased with increasing DII scores. IL-10 concentrations decreased in pregnant women whose DII scores increased between the second and third trimesters (*p* = 0.011). Thiamin and vitamin C were negatively correlated with MCP-1 (β = −0.879, and β = −0.003) and IL-6 (β = −0.602, and β = −0.002) levels in the third trimester. In conclusion, the DII score had a U-shaped association with cytokine levels during the third trimester. Changes in DII scores between the second and third trimesters of pregnancy were correlated with cytokine levels during the third trimester.

## 1. Introduction

Maternal inflammation levels during pregnancy play an important role in fetal development, and cytokines such as interleukin 1β (IL-1β), IL-6, and monocyte chemoattractant protein-1 (MCP-1) change with the progression of pregnancy [1]. An exacerbated inflammatory response in pregnant women is not beneficial because it can affect the health of both the pregnant woman and the fetus [2,3]. A large number of studies have shown that high cytokine levels are associated with pregnancy complications (e.g., gestational diabetes, C-reactive protein (CRP), and IL-6 levels [4,5], gestational hypertension and CRP levels [6], preeclampsia and tumor necrosis factor-α (TNF-α) levels, IL-6 levels, IL-8 levels and hematological parameters of systemic inflammation (NLR/MPV) [7,8]) and adverse birth outcomes (e.g., low birth weight infants and CRP levels [9], premature delivery and CRP levels [10], prolonged pregnancy and IL-8 levels [11]). Maternal immune system diseases and serum IL-1β, IL-6, MCP-1, and TNF-αlevels can even affect the neurodevelopment of offspring [12,13,14,15,16]. TNF-α, CRP, IL-1β, and IL-6 levels are common indicators of inflammation [17,18,19]; IL-10 is one of the main anti-inflammatory markers in the human body [20]. MCP-1 and IL-8 are associated with the health of mothers and their offspring [11,15]. Therefore, seven cytokines (TNF-α, IL-1β, CRP, MCP-1, IL-6, IL-8, and IL-10) were included in this research.

Dietary components are closely related to chronic inflammation in the body [21,22,23,24]. Studies have shown that dietary intake and dietary patterns may differentially affect systemic inflammation [25,26,27]. The dietary inflammatory index (DII) was developed as a tool to measure the inflammatory effects of diet on the human body [23,28]. Previously, the DII has been used to predict the levels of certain cytokines in the general adult population in different countries (e.g., Japan [29], Korea [30], and the USA [31,32]). Currently, it is widely used to investigate the impact of diet-induced inflammation on cancer [33,34], diabetes [35,36], hypertension [37], psychological disorders [38], metabolic syndrome [39,40], and other diseases. Among the types of validations performed worldwide, only a few studies have focused on the relationship between the DII score and cytokine levels in pregnant women in China, and these studies only investigated the dietary status in a certain trimester of pregnancy and a single cytokine [9,41,42]. Previous studies have shown that the DII score is correlated with health problems in offspring [43,44,45], but evidence of the relationship between the DII score and cytokine levels in pregnant women is limited. One study investigated the relationship between the DII scores of pregnant women and three cytokine levels in the first, second, and third trimesters, but different relationships were found among the different trimesters and groups [46]. As pregnancy is considered to be a natural inflammatory state [47,48,49], molecular immune response pathways are different from those in nonpregnant adults, and it is unclear whether the DII score affects cytokine levels in pregnant women in the same way that it affects nonpregnant adults. Moreover, cytokine concentrations fluctuate across all three trimesters of pregnancy [1], which means that the inflammatory state changes during pregnancy.

In addition, evidence showed that a high serum CRP level in the second trimester was associated with an increased risk of delivering preterm [50]. In the third trimester, maternal plasma cytokine levels were also correlated with offspring neurodevelopment [51]. Moreover, Chinese pregnant women are affected by traditional dietary habits and will increase their intake of various nutrients during the second and third trimesters. However, studies on the effects of the dietary inflammatory potential on cytokine levels in pregnant women during the second and third trimesters in China are limited. Thus, the influence of the DII of the Chinese dietary pattern on cytokine levels during the second and third trimesters in Chinese pregnant women needs further investigation.

In this study, the diet information of pregnant women was collected during the second and third trimesters, and the DII score and cytokine levels were measured to describe the DII level during the second and third trimesters in northern China and to analyze the relationship between the DII score and cytokine levels in pregnant women during the second and third trimesters.

## 2. Materials and Methods

### 2.1. Study Design and Participants

Participants were selected from the Tianjin Maternal and Child Health Education and Service Cohort (TMCHESC). This was a prospective birth cohort study to explore the environmental risk factors for neurodevelopmental disorders in children in Tianjin, China. Pregnant women were recruited at their first prenatal visit at the Women and Children’s Health Center in Hebei District and Heping District, Tianjin Province, China, from February 2018 to October 2021. The inclusion criteria were as follows: (1) women who were within 13 weeks of pregnancy; (2) women aged > 18 years; (3) women without communication problems; (4) women planning to deliver in Tianjin, with no plan to leave Tianjin within four years; and (5) women without a history of COVID-19 infection.

During the follow-up until delivery, a total of 175 pregnant women were included in this study after excluding those who were lost to follow-up during pregnancy, those who refused diet evaluation during pregnancy, those with autoimmune diseases or anti-inflammatory medication use, those who did not complete the nutritional supplementation status questionnaire, and those with a total energy intake <600 or >3500 kcal/d. Seventy-four pregnant women participated in both the second- and third-trimester surveys. Eighty-five pregnant women completed follow-up only in the second trimester; sixteen women completed follow-up only in the third trimester (Figure 1).

The study was carried out following the guidelines of the Declaration of Helsinki, and all procedures involving human subjects were approved by the Ethics Review Committee of Tianjin Medical University (TMUhMEC2017020). All participants were presented with the contents of the TMCHESC and provided written informed consent upon recruitment in this study.

### 2.2. Questionnaires

The following information was collected in the first trimester: age, last menstrual period, pre-pregnancy weight, pre-pregnancy height, education level, annual household income, smoking status, secondhand smoke exposure, alcohol consumption, autoimmune diseases, history of chronic inflammatory diseases, gravidity, and parity. The following information was collected during the second- and third-trimester follow-ups: physical activity level, smoking status, alcohol consumption, any complications, and the use of nutritional supplements (such as calcium, iron, folic acid, and vitamins). The following information was collected from medical records: gestational week of delivery and sex of the fetus.

### 2.3. Dietary Assessment

In this study, three day 24-h food records (two weekdays and one weekend), including food and nutritional supplements, were collected face-to-face from each participant in the second and third trimesters by a trained dietitian referring to an image-based dietary assessment method [52] to estimate their nutrient intake during a specific pregnancy. For those who went through two visits in the second and third trimesters, the dietary intake was measured in both trimesters. To ensure accuracy, the pregnant women were instructed on how to record their dietary intake for three days (two working days and one weekend day) by the researchers a few days before the survey [53]. Photographs and text records were allowed. The consumption of tea, coffee, and alcohol in the last week was investigated using a questionnaire. Average daily energy and other nutrient intakes were calculated by the Yingyangjisuanqi v2.7.6.10 computer program adapted to the Chinese diet and the Chinese Food Composition Tables (6th edition) [54].

### 2.4. DII Assessment

The original DII was created by Cavicchia using CRP validation in longitudinal data from the Seasonal Variation of Cholesterol Levels Study (SEASONS) in 2009 [28]. The goal was to provide a personal dietary tool that continuously classifies minimum inflammation to maximum inflammation. It was later further standardized by Shivappa [23]. Ultimately, they screened 6500 articles on the effects of dietary parameters on six inflammatory markers to develop the DII scoring algorithm based on whether each dietary parameter had an increase (+1), decrease (−1), or no (0) effect on inflammatory biomarkers. Six inflammatory markers are included: DII: IL-1β, IL-4, IL-6, IL-10, TNF-α and CRP. The calculation of the DII score involved 11 food consumption datasets and 45 food parameters from various countries around the world. The DII score reflects both a robust literature base and the standardization of individual intake to global reference values.

Due to some food parameters not being available in the latest version of Chinese Food Composition Tables, and based on the literature and local dietary habits, twenty-nine of the forty-five food parameters were selected from the DII database for calculation: energy, carbohydrates, protein, fat, alcohol, fiber, cholesterol, saturated fatty acids (SFAs), monounsaturated fatty acids (MUFAs), polyunsaturated fatty acids (PUFAs), niacin, riboflavin, vitamin B12, vitamin B6, vitamin A, β-carotene, vitamin C, vitamin D, vitamin E, folic acid, iron (Fe), magnesium (Mg), zinc (Zn), selenium (Se), caffeine, onion, garlic, ginger, and tea.

Each nutrient intake was converted to a z score using global means and standard deviations for each food parameter provided in the DII database. Then, the z scores were converted into central percentiles. Nutrient parameter-specific DII scores were calculated by multiplying the central percentiles by the respective inflammatory effect parameter. Finally, the nutrient parameter-specific DII scores of the same person were summed to calculate her total DII score. Higher scores represent pro-inflammation, while lower scores represent anti-inflammation.

### 2.5. Blood Samples

Fasting blood samples were collected from participants at two time points during pregnancy (second trimester: 19.39 ± 3.29 weeks, third trimester: 29.57 ± 2.17 weeks) by coagulation-promoting tubes. Peripheral venous blood was drawn (5 mL) by registered nurses in the early morning. The tubes were placed upright on the test tube rack for 60 min. The serum was separated after centrifugation (3000× *g*) for 15 min at 4 °C after checking that the coagulation was complete. Then, it was stored in an ultralow temperature freezer at −80 °C until determination.

### 2.6. Biochemical Measurements

IL-1β, IL-6, IL-8, IL-10, and MCP-1 concentrations were measured using a High Sensitivity ELISA kit (Liankeshengwu, China), and TNF-α and CRP concentrations were measured using an ELISA kit (Elabscience, China) according to the manufacturer’s guidelines. The correlation coefficient between the standard and expected concentrations was greater than 0.999, with good accuracy. The standard provided by the manufacturer was diluted to obtain the standard curve for each test. Results from each experiment were obtained from this curve.

### 2.7. Statistical Analysis

Continuous variables are presented as the mean ± standard deviation (SD) or Median (upper quartile, lower quartile); categorical variables are presented as frequencies of occurrence and percentages. Multivariable generalized linear models were used to investigate influencing factors of DII values during the second (*n* = 159) and third trimesters (*n* = 90) of pregnancy, respectively. After adjusting for energy intake, age, education level, annual household income, and BMI before pregnancy [55,56], restricted cubic splines were used to explore the relationship between the levels of each cytokine and the DII values in the second (*n* = 159) and third (*n* = 90) trimesters. Multivariable generalized linear models were used to analyze the relationship between nutrients and cytokines in the second and third trimesters.

Subgroup analysis was conducted using data from 74 pregnant women who participated in both the second and third trimesters. The concentrations of cytokines were log-transformed to a base of 10 because they were nonnormally distributed. Differences in DII scores and cytokine levels between the second and third trimesters were analyzed by a paired *t* test and Wilcoxon signed-rank test. Differences in DII values and cytokine levels between the second and third trimesters (T3-T2) of each participant and the percentiles of these differences were calculated. Then, the participants were classified into three different groups (<35th percentile, 35th–65th percentile, and ≥65th percentile) based on the percentile of their DII difference (T3-T2). One-way ANOVA with LSD adjustment was used to investigate cytokine differences among the three groups after adjusting for second-trimester cytokine concentrations and energy intake in the second and third trimesters.

All statistical analyses were performed with SPSS 25.0 and R3.4.3. A two-sided *p* value < 0.05 was considered statistically significant.

## 3. Results

### 3.1. Participant Baseline Characteristics

The mean age of the 175 subjects was 30.86 ± 3.47 years. The prepregnancy body mass index (BMI) was 22.23 ± 3.59 kg/m^2^. The difference in demographic information between the 74 pregnant women followed in both the second and third trimesters and the other pregnant women were not statistically significant.

The mean DII scores were −0.07 ± 1.65 and 0.06 ± 1.65 in the second and third trimesters, respectively (Table 1, Appendix A).

### 3.2. Effect of Basic Characteristics on DII Scores

After controlling for other variables in the model, age and education level were influential factors for the second-trimester DII score; gestational age and household income were influential factors for the third-trimester DII score.

In the second trimester, for each year of age, the DII score decreased by 0.09 (95% CI: −0.17, −0.02). For pregnant women with low and moderate education, the DII scores were 1.28 and 1.16 higher than those with high education, respectively.

In the third trimester, for every week of gestational age, the DII score decreased by 0.17 (95% CI: −0.32, −0.02). For pregnant women with low and moderate annual household incomes, the DII scores were 2.50 and 1.11 higher than those with high annual household incomes, respectively. (Table 2).

### 3.3. Relationship between the DII Score and Cytokine Levels during Pregnancy

In the third trimester, a U-shaped association was found between IL-1β levels (*p* = 0.039) and MCP-1 levels (*p* = 0.035) and DII scores (Figure 2, Appendix A). They both decreased with increasing DII scores and then increased.

### 3.4. Relationship between Nutrients/Food Components and Cytokine Levels during Pregnancy

After adjusting for confounding variables in the model, there are nine kinds of nutrients/food components influent the third-trimester cytokine levels; fifteen kinds of nutrients/food components influent the third-trimester cytokine levels.

In the second trimester, as total fat intake increased, the concentration of IL-8 increased. In the third trimester, as SFAs and PUFAs intake increased, the IL-8 level decreased; as carbohydrates and fiber intake increased, the IL-8 level increased. When niacin intake decreased and Vitamin E intake increased, the TNF-α level increased. As protein intake decreased and Vitamin B6 intake increased, the CRP level increased. As thiamin and VC intake increased, MCP-1 and IL-6 levels decreased (Table 3).

### 3.5. Cytokine Levels in Women with Repeated Trimester Measures

There were no statistically significant differences in DII scores between the second- and third-trimester pregnant women (*p* = 0.633).

In the 74 pregnant women with repeated measurements, the concentrations of IL-6 were higher in the third trimester than in the second trimester (1.32 vs. 0.61 pg/mL), and the difference was statistically significant (*p* < 0.001). Only 14 participants decreased IL6 concentrations in the third trimester. The concentrations of the other six cytokines (TNF-α, IL-1β, CRP, MCP-1, IL-8, and IL-10) were not significantly different between the second and third trimesters (Table 4).

### 3.6. Changes in DII Scores and Cytokine Levels during Pregnancy

The concentrations of IL-10 in the third trimester correlated with the DII difference between the third and second trimesters (T3-T2) (*p* = 0.011), and the concentrations of IL-10 were lower in the ≥P65 group than in the P35-P65 group (Figure 3, Appendix A).

## 4. Discussion

In this study, we first investigated the relationships between DII scores and cytokine levels in pregnant women in the second and third trimesters. In the third trimester, there was a U-shaped association between DII scores and cytokine levels. As DII scores increased, IL-1β and MCP-1 levels decreased and then increased. In pregnant women with repeated measures, the concentration of IL-10 in the third trimester correlated with the DII score difference between the third and second trimesters.

In the present study, the mean DII scores in the second and third trimesters were approximately 0, which is similar to the findings in other provinces in China [41,57,58]. The DII scores of pregnant women in Poland [46], adults in Europe [59], and Caucasians [60] were all lower than the general scores of pregnant women in China. However, the DII scores of pregnant women in Iran [61] were higher than those of pregnant women in China. Therefore, the DII scores of Chinese pregnant women were moderate compared to those of women worldwide.

We found that pregnant women with higher DII scores in the third trimester had lower levels of IL-10 in the third trimester than in the second trimester. This seems to suggest that changes in diet may affect cytokine concentrations in pregnant women. Similarly, in previous studies with nonpregnant women, dietary intervention decreased the concentrations of anti-inflammatory cytokines [62,63,64]. IL-10 is a nonredundant cytokine that limits the inflammatory response [20]. This cytokine plays a key regulatory role in immune tolerance at the placental barrier during pregnancy to balance proinflammatory and anti-inflammatory signals and promote better pregnancy outcomes [65,66]. The third trimester is the maternal proinflammatory phase, and low levels of IL-10 due to diet may induce adverse pregnancy outcomes such as miscarriage, recurrent miscarriage, preterm delivery, and preeclampsia [66,67,68].

In the third trimester, IL-1β and MCP-1 levels showed a U-shaped association with increasing DII scores. These factors both decreased with the DII score and then increased. Most studies have shown that an anti-inflammatory diet is associated with increases in anti-inflammatory cytokines and decreases in proinflammatory cytokines in the general population [69,70,71]. Mechanistically, dietary changes may alter the inflammatory state due to fiber and other antioxidant substances [72,73]. However, many studies with pregnant women have not identified this correlation [46,74,75,76]. Pieczynska showed that in the first trimester, the relationship between CRP concentrations and DII scores differed in the groups of DII scores < the median and DII scores > the median [46]. We also found a U-shaped relationship between the DII score and cytokine levels. The main reason for this finding is that the inflammatory status of pregnancy itself changes during different periods [1,77], and pregnant women have a buffer effect on activities that may change their inflammatory status during pregnancy to maintain a normal pregnancy [78]. However, the mechanism involved is unknown. Some relationships between DII and cytokines were not statistically significant. Possible explanations could be: due to different dietary habits or populations, the types or quantities of nutrients taken by pregnant women in China may different from Western countries where DII was developed.

We found large differences in the effect of different foods on cytokines between the second and third trimesters, which is probably because of inconsistent inflammatory status in pregnancy. Pregnant women were in an anti-inflammatory state in the second trimester and a proinflammatory state in the third trimester [1]. Therefore, we reasonably suspect that sensitivity to different nutrients is inconsistent across pregnancy periods.

In this study, SFAs, PUFAs, protein, thiamin, vitamin C, and niacin showed anti-inflammatory effects. The underlying molecular mechanisms are complex. N-3/n-6 PUFAs regulate hypothalamic fatty acid profiles, gene expression, and insulin signaling, and neurons decrease inflammation in response to hypothalamic synergy [79]. Thiamine and niacin inhibit oxidative stress-induced (nuclear factor kappa-B (NF-κB) activation [80,81,82,83], and vitamin C regulates the activity of NADPH oxidase (NOX) to inhibit proinflammatory cytokine production [84]. However, it should be noted that SFAs facilitate the activation of TLR4 and promote the CD14-TLR4-MD2 complex inflammatory pathway, which induces proinflammatory effects [79,85]. However, there may be synergistic or antagonistic effects between ingested nutrients, causing the conflicting results of this study [86,87]. It is important to note that this study did not further classify these macronutrients, carbohydrates, or proteins, thus reducing the interpretability of these findings. Nutrients involved in energy metabolism may influence the levels of inflammatory factors by affecting BMI and other immune cells [88]. Carbohydrates include complex carbohydrates and simple carbohydrates, which are derived from whole, plant-based foods and ultra-processed foods, respectively [89]. Generally, complex carbohydrates are considered anti-inflammatory; simple carbohydrates are considered pro-inflammatory [89].

Another important finding is that fiber [90], vitamin E [91], vitamin B6 [92], and Se [93], which are usually considered anti-inflammatory nutrients, mediate the proinflammatory effects required by pregnant women in the third trimester. The demand for these nutrients is conducive to health during pregnancy due to fetal growth and development [94]. Therefore, we believe that the reason is that the intake of these nutrients has not yet satisfied the demands of pregnant women (Appendix A). We found that the intake of certain dietary nutrients is associated with cytokine levels in pregnant women. We believed that diet may influence maternal and offspring health by influencing levels of inflammation. Therefore, we should consider the effect of nutrients/food ingredients on inflammation when providing dietary advice to or performing evaluations for pregnant women. Further studies are needed to investigate the causal relationship and make recommendations in the future.

There were several strengths in this study. First, we revealed the relationship between DII scores and cytokine concentrations in the second and third trimesters, respectively. In addition, the effects of nutrients/food components on cytokine levels in pregnant women during different trimesters were explored. Crucially, we found a statistically significant difference in cytokine levels between pregnant women with elevated third-trimester DII scores compared to those with unchanged DII scores. However, our study still has some limitations. (1) The 24-h food records that we used may not represent the entire condition during each trimester. (2) Each pregnant woman received dietary guidance during the dietary survey process, and their food choices may have changed. (3) The subjects included in this study were all healthy before pregnancy and did not have extreme inflammatory conditions, so they do not represent all pregnant women.

## 5. Conclusions

In conclusion, there was no difference in DII scores between second- and third-trimester pregnant women in China. There is a U-shaped relationship between the DII score and IL-1β and MCP-1 levels, and changes in the DII score in the third trimester relative to the second trimester may change IL-10 levels in the third trimester. Changes in the DII score between the third trimester and the second trimester correlated with cytokine concentrations in the third trimester. Future studies should explore the mechanistic relationship between maternal DII scores and cytokine levels in different trimesters to develop nutritional guidelines during pregnancy that are more helpful for maintaining maternal and fetal health. In addition, more studies are needed to explore the effects of good long-term dietary habits on maternal, fetal, and offspring postnatal health.

## Figures and Tables

**Figure 1 nutrients-15-00194-f001:**
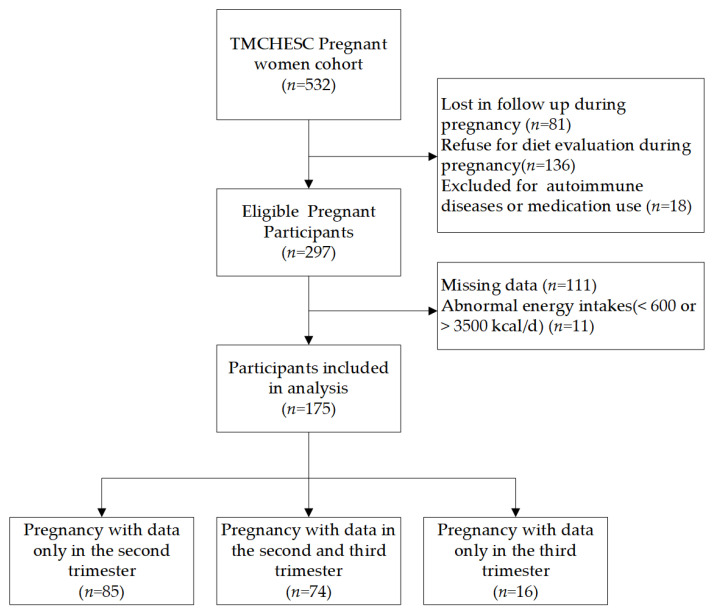
Flow chart of sample selection in this study.

**Figure 2 nutrients-15-00194-f002:**
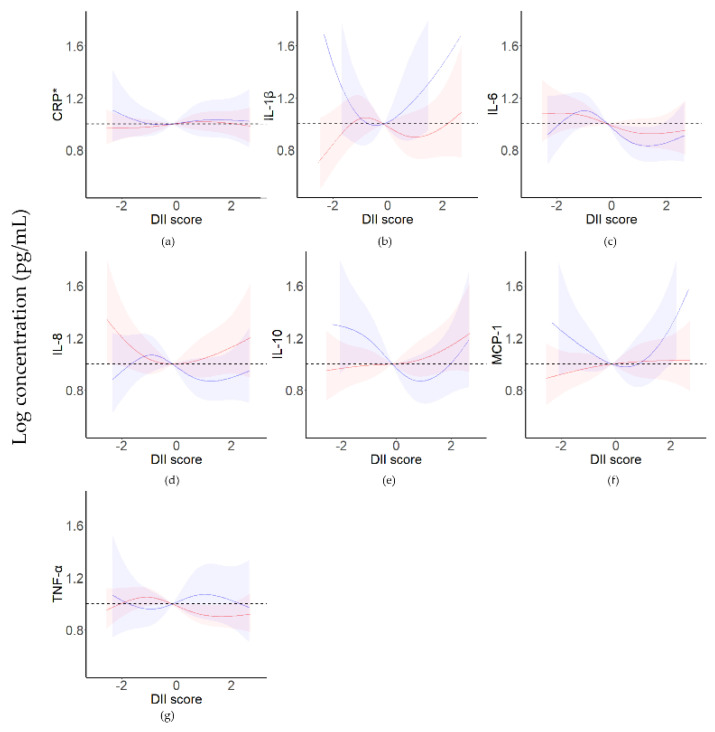
Restrictive cubic spline analysis of cytokine levels and DII scores for pregnant women in the second and third trimesters. The red line represents pregnant women in the second trimester (*n* = 159); the blue line represents pregnant women in the third trimester (*n* = 90). CRP, C−reactive protein; IL, interleukin; MCP−1, monocyte chemoattractant protein−1; TNF−α, tumor necrosis factor−α. (**a**) CRP, (**b**) IL−1β, (**c**) IL−6, (**d**) IL−8, (**e**) IL−10, (**f**) MCP−1, (**g**) TNF−α. Energy intake, age, education level, annual household income, and BMI were used as covariates. * CRP level is the logarithm of the cytokine level (ng/mL).

**Figure 3 nutrients-15-00194-f003:**
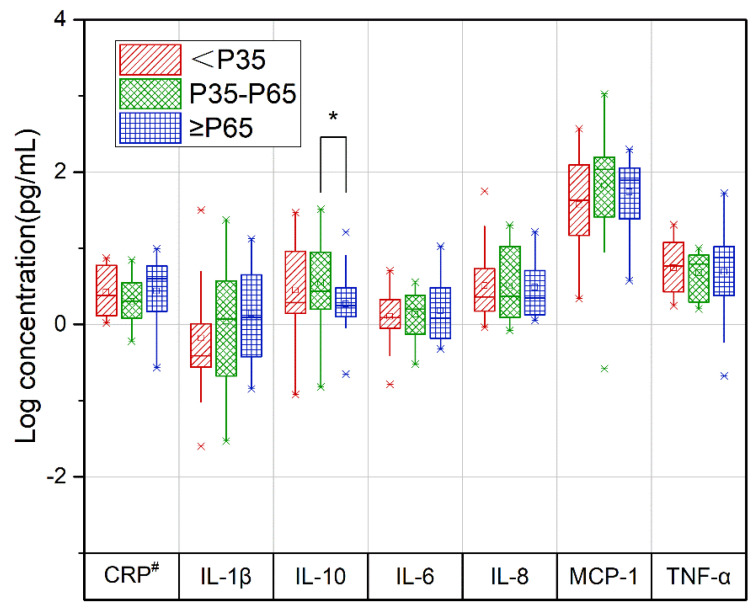
Relationship between the change in repeated measurements of the DII and cytokine levels in pregnant women during the second and third trimesters of pregnancy. Red slashes represent the <P35 group, green diamonds represent the P35-P65 group, and blue squares represent the ≥P65 group. CRP, C−reactive protein; IL−1β, interleukin 1β; IL−6, interleukin 6; IL−8, interleukin 8; IL−10, interleukin 10; MCP−1, monocyte chemoattractant protein−1; T2, second trimester; T3, third trimester; TNF−α, serum tumor necrosis factor−α. Concentrations of cytokines in the second trimester and energy intake in the second and third trimesters were treated as covariates. Data from the 74 pregnant women who completed the second- and third-trimester follow-ups. * *p* value for the comparison between P35-P65 and ≥P65 groups, α = 0.017. ^#^ CRP level is the logarithm of the cytokine level (ng/mL).

**Table 1 nutrients-15-00194-t001:** Characteristics and DII scores of participants (*n* = 175) ^#^.

	Participants
Age (years)	30.86 ± 3.47
Prepregnancy BMI (kg/m^2^)	22.23 ± 3.59
Education level (years)	
≤12	19 (10.86%)
12~15	139 (79.43%)
≥15	17 (9.71%)
Annual household income (CNY)	
≤120,000	15 (8.57%)
120,000~240,000	132 (75.43%)
≥240,000	28 (16.00%)
Smoking during the prepregnancy period	
No	165 (94.29%)
Yes	10 (5.71%)
Prepregnancy secondhand smoke exposure	
No	125 (71.43%)
Yes	50 (28.57%)
Secondhand smoke exposure in the second trimester *	
No	111 (69.81%)
Yes	48 (30.19%)
Secondhand smoke exposure in the third trimester **	
No	71 (78.89%)
Yes	19 (21.11%)
Drinking during prepregnancy period	
No	171 (97.71%)
Yes	4 (2.29%)
Second Trimester physical activity * (MET h/day)	
Resting	14.13 (10.82, 17.40)
Light	9.38 (5.00, 13.00)
Moderate	10.50 (7.50, 14.00)
Heavy	0.00 (0.00, 0.00)
Third Trimester physical activity ** (MET h/day)	
Resting	12.92 (10.08, 16.00)
Light	10.70 (5.50, 14.19)
Moderate	11.65 (7.50, 15.13)
Heavy	0.00 (0.00, 0.00)
DII score	
Second Trimester *	−0.07 ± 1.65
Third Trimester **	0.06 ± 1.65

BMI, body mass index; CNY, Chinese yuan; DII, dietary inflammatory index; MET, metabolic equivalents; T2, second trimester; T3, third trimester; ^#^ Continuous variables are presented as the mean ± SD or median (upper quartile, lower quartile), and categorical variables are shown as *n* (%); * Data from 159 pregnant women who completed the second-trimester follow-up; ** Data from 90 pregnant women who completed the third-trimester follow-up.

**Table 2 nutrients-15-00194-t002:** Multivariate general linear model analysis of demographic factors and DII scores.

	DII Score in the Second Trimester *	*p* _1_	DII Score in the Third Trimester **	*p* _2_
	B (95% CI)		B (95% CI)	
Age (years)	−0.09 (−0.17, −0.02)	0.014	−0.09 (−0.18, 0.00)	0.052
Prepregnancy BMI (kg/m^2^)	−0.02 (−0.12, 0.09)	0.763	−0.09 (−0.24, 0.06)	0.263
Gestational age (weeks)	0.01 (−0.07, 0.08)	0.872	−0.17 (−0.32, −0.02)	0.022
Pregnancy BMI (kg/m^2^)	0.03 (−0.07, 0.12)	0.562	0.13 (−0.04, 0.30)	0.124
Physical activity (MET h/day)				
Resting	−0.19 (−0.73, 0.35)	0.486	−0.02 (−0.30, 0.26)	0.890
Light	−0.13 (−0.47, 0.20)	0.439	0.00 (−0.17, 0.17)	0.991
Moderate	−0.03 (−0.17, 0.11)	0.627	0.01 (−0.09, 0.11)	0.823
Heavy	−0.33 (−0.79, 0.13)	0.163	0.02 (−0.18, 0.21)	0.861
Education level (years)				
≤12	1.28 (0.07, 2.49)	0.038	−0.14 (−0.15, 1.23)	0.846
12~15	1.16 (0.24, 2.09)	0.014	0.13 (−0.81, 1.08)	0.781
≥15	0		0	
Annual household income (CNY)				
≤120,000	0.18 (−0.87, 1.23)	0.733	2.50 (1.09, 3.91)	0.001
120,000~240,000	0.13 (−0.54, 0.81)	0.698	1.11 (0.05, 2.16)	0.039
≥240,000	0		0	
Smoking during the prepregnancy period		0.878		0.882
Yes	0.09 (−1.01, 1.18)		−0.18 (−2.55, 2.19)	
No	0		0	
Prepregnancy secondhand smoke exposure		0.715		0.133
Yes	0.11 (−0.48, 0.71)		0.57 (−0.17, 1.32)	
No	0		0	
Drinking during the Prepregnancy period		0.902		0.088
Yes	0.12 (−1.80, 2.04)		1.80 (−0.26, 3.86)	
No	0		0	
Pregnancy secondhand smoke exposure		0.738		0.558
Yes	0.10 (−0.48, 0.68)		−0.26 (−1.13, 0.61)	
No	0		0	

BMI, body mass index; CNY, Chinese yuan; DII, dietary inflammatory index; MET, metabolic equivalents; * Data from 159 pregnant women who completed the second-trimester follow-up; ** Data from 90 pregnant women who completed the third-trimester follow-up.

**Table 3 nutrients-15-00194-t003:** Multivariate general linear model analysis of cytokine levels and major nutrients/food components (B).

	T2 ^#^							T3 ^$^						
	TNF-α	IL-1β	CRP	MCP-1	IL-6	IL-8	IL-10	TNF-α	IL-1β	CRP	MCP-1	IL-6	IL-8	IL-10
β-carotenes (mg)	−0.006	−0.063	0.001	−0.040	0.077 *	0.068	−0.004	0.110	0.197	−0.028	0.002	0.014	−0.022	0.103
SFAs (kg)	−0.041	4.003	−5.602	−4.043	−2.126	−2.659	5.746	15.615	3.441	−5.808	3.807	8.895	−57.744 *	−15.385
MUFAs (kg)	−2.037	13.337	−0.049	0.998	1.372	−11.778	11.632	−1.183	−39.581	9.963	−1.371	31.757	75.872 *	−6.024
PUFAs (kg)	−0.384	−7.785	2.179	−2.525	−5.815	−1.868	−6.117	−17.129	24.640	−3.917	−9.323	−19.402	−34.231 *	6.757
Protein (kg)	−2.823	4.728	−1.916	2.682	−2.126	−0.797	4.745	3.059	14.157	−9.174 *	1.361	−4.151	−2.407	8.992
Total fat (kg)	1.373	−0.539	1.711	4.712	−1.120	9.784 *	−3.679	3.523	2.921	1.079	2.379	−5.264	−3.503	−3.515
Carbohydrates (kg)	−0.519	−1.418	0.095	−1.037	0.289	−0.750	−0.567	0.074	−0.851	1.018	−1.298	1.335	2.454 *	−3.707
Fiber (kg)	5.837	−10.707	−2.158	−4.822	3.217	−12.347	−9.584	−3.216	2.346	5.420	24.027	22.867	31.786 *	12.533
Cholesterol (g)	−0.111	−0.302	0.182	−0.508 *	0.191	−0.315	−0.049	−0.497	−0.313	0.102	−0.472	0.054	−0.081	−0.668
Vitamin A (RE × 1000)	0.043	0.127	0.012	0.133 *	0.021	−0.002	−0.022	−0.095	0.156	−0.143	0.053	0.094	0.044	0.052
Vitamin D (mg)	1.400	1.781	1.393	−0.374	0.167	2.793	2.447	−5.432	−16.065	−5.561	−8.376	−1.197	−9.779	−25.791 *
Vitamin E (g)	3.457	−9.954	−0.636	−2.835	3.053	11.057	5.194	20.098 *	−18.074	1.257	5.287	19.495 *	11.286	−30.201 *
Thiamin (mg)	−0.136	−0.168	−0.076	0.108	0.243	0.074	−0.283	−0.384	−0.123	0.076	−0.879 *	−0.602 *	−0.478	0.393
Riboflavin (mg)	−0.021	0.064	−0.044	−0.034	−0.043	−0.067	0.116 *	0.163	−0.175	−0.083	0.064	−0.176	−0.061	0.041
Vitamin B6 (mg)	0.078	−0.085	0.213	−0.174	−0.059	−0.065	0.200	0.532	−0.440	0.669 *	−0.349	0.406	0.511	−0.391
Vitamin B12 (μg)	−0.010	0.070	−0.099	0.047	−0.011	−0.020	−0.147	−0.508	0.445	−0.362	0.532	0.061	−0.125	0.207
IL-6 (g)	−0.919	1.323	−0.487	0.806	−0.418	−0.222	0.685	1.273	−1.364	0.159	−2.888 *	−1.794 *	−0.433	−1.250
Folic acid (mg)	0.150	−0.339	0.078	−0.358	−0.151	−0.222	−0.102	0.216	−0.237	−0.112	0.363	−0.325	−0.434	−0.101
Niacin (g)	4.788	9.442	−0.519	7.310	11.739	−9.219	6.653	−32.335 *	−19.130	−0.544	12.451	−2.881	−6.040	5.186
Mg (g)	0.397	1.986	0.307	0.103	−0.684	−0.879	0.222	−2.056	−0.697	−0.292	−0.353	−0.789	−0.827	2.231
Fe (g)	−1.462	2.835	−1.609	2.726	1.523	−1.368	2.683	15.640 *	5.617	0.202	−0.957	2.101	5.491	11.353 *
Zn (g)	13.237	−60.299 *	11.816	−26.479	−13.029	26.992	−26.969	−30.878	−1.211	−8.921	18.601	−45.600	−38.110	−69.354 *
Se (mg)	1.206	−1.458	0.776	4.212	1.642	3.717	−3.438	7.687	4.716	6.643	6.677	10.970 *	6.238	11.536

CRP, C-reactive protein; Fe, iron; IL-1β, interleukin 1β; MCP-1, monocyte chemoattractant protein-1; Mg, magnesium; MUFAs, monounsaturated fatty acids; PUFAs, polyunsaturated fatty acids; Se, selenium; SFAs, saturated fatty acids; T2, second trimester; T3, third trimester; TNF-α, tumor necrosis factor-α; Zn, zinc; ^#^ Data from 159 pregnant women who completed the second-trimester follow-up; ^$^ Data from 90 pregnant women who completed the third-trimester follow-up; * *p* < 0.05.

**Table 4 nutrients-15-00194-t004:** Concentrations of cytokines in pregnant women with repeated measurements (*n* = 74).

	T2	T3	T3 < T2 *n* (%)	*p* *
DII score	−0.04 ± 1.63	−0.15 ± 1.56	34 (45.95%)	0.633
TNF-α (pg/mL)	7.41 (4.15, 9.55)	6.66 (2.46, 9.10)	32 (43.24%)	0.066
IL-1β (pg/mL)	1.11 (0.37, 5.88)	0.83 (0.31, 3.95)	26 (35.14%)	0.901
CRP (ng/mL)	3.10 (1.81, 4.75)	2.49 (1.30, 4.59)	36 (48.65%)	0.214
MCP-1 (pg/mL)	77.24 (21.68, 146.77)	66.04 (19.94, 133.18)	38 (51.35%)	0.353
IL-6 (pg/mL)	0.61 (0.35, 0.98)	1.32 (0.70, 2.69)	14 (18.92%)	<0.001
IL-8 (pg/mL)	1.92 (1.29, 8.90)	2.26 (1.29, 8.37)	26 (35.14%)	0.940
IL-10 (pg/mL)	3.09 (1.45, 10.23)	1.96 (1.35, 7.47)	29 (39.19%)	0.714

CRP, C-reactive protein; IL-1β, interleukin 1β; MCP-1, monocyte chemoattractant protein-1; T2, second trimester; T3, third trimester; TNF-α, tumor necrosis factor-α; * *p* values for paired *t* test for DII scores and Wilcoxon signed-rank test for cytokines in the second and third trimesters for pregnant women who participated in both the second and third trimesters (*n* = 74).

## Data Availability

The datasets generated and analyzed during this study are not publicly available to protect the privacy of the participants. The data are available from the Department of Maternal, Child, and Adolescent Health at the School of Public Health of Tianjin Medical University and can be obtained from the corresponding author (email: zhangxinty06@163.com) upon reasonable request.

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
