# Peer review of "Relationship between the Dietary Inflammatory Index Score and Cytokine Levels in Chinese Pregnant Women during the Second and Third Trimesters"

_nutrients, 2022, doi:10.3390/nu15010194_

Round 1

Reviewer 1 Report (Previous Reviewer 2)

Very nice work. 

Author Response

Thank you for your careful reading and review of this manuscript. It is a great honor to receive your approval of this study, which is a great encouragement to us. We have further polished the manuscript and made changes for better presentation.

Reviewer 2 Report (New Reviewer)

The authors explored relationships between dietary intake of food and nutrients and their possible connection with inflammatory markers IL-1B, IL-6, IL-8, IL-10, CRP and MCP-1. 

Major suggestions

There are several questions considering study design. The authors have two groups of pregnant women one with 159 women and the other with 90. Why they stopped recruitment of women in the third trimester of pregnancy before they had two groups with comparable number of participants? The second question was it scientifically correct to collect data for 29 instead of 45 parameters and 5 foods instead of 11 foods? Could it be additional factor that led to no correlations found between calculated DII score and analyzed inflammation parameters? According to obtained data whether nutrition is an important factor that influence inflammatory parameters status during second and third trimesters of pregnancy in Chinese women? 

Results section

It seems that the results for DII scores are expressed as mean +- standard and stdev values are very high. Is the normality of distribution of these data tested? 

The correlation between age and DII score is not negative if the lower the age was, the lower the DII scores. It could be useful to add the coefficients of correlations could be in Table 2 as there is not many texts in results section about obtained correlations and so a reader cannot get in sighed if these correlations are direct or inverse?

I suggest to the authors to rewrite lines 241-249, description of data presented in Table 3 using the terms inversely or directly associated or something similar. I am not a statistician, but I think that some additional information could be provided when b is zero. Is it possible that correlation with coefficient 0.000 is statistically significant. Could the author provide some explanations for big differences for results between T2 and T3 presented in Table 3. They could be added in discussion section, too.

The data about the number of women those inflammation parameters are increased or decreased comparing trimesters could be added (comparing 74 subjects, Table 4).

Line 408-410 Does the changes in the DII score correlate with cytokine concentrations? There is no change in DII scores between trimesters, and it is noticed only for IL-10. 

Minor points

Line 66 term normal adults should be replaced 

Line 119 3-day food records it is usually called 24h food record

Line 379 We boldly reason should be replaced with some more appropriate term (pointed out or similar)

Author Response

Thanks very much for taking your time to review this manuscript. I really appreciate all your comments and suggestions. Please find our responses in below and my revisions in the re-submitted files.

Round 2

Reviewer 2 Report (New Reviewer)

In their response document the authors provided answers to all questions that were addressed to them. The quality of the manuscript was improved with added corrections in text. The obtained results were explained in more details. The statistical significance of results is better described. The authors comment more carefully obtained results and so they were in better correlations with conclusions that were stated.

From all above mentioned, the current form of manuscript could be accepted for publication by my opinion.

This manuscript is a resubmission of an earlier submission. The following is a list of the peer review reports and author responses from that submission.

Round 1

Reviewer 1 Report

This manuscript presents data related to elaborating the relationship between diet-associated inflammation and changing cytokine levels in the second and third trimester of pregnancy. It is not clear if this is a prospective study of the same women given the different numbers of participants in each trimester (as in Table 1) and the below should also be considered. 

1.     In the introduction the authors note a relationship between high cytokine levels and pregnancy complications and then cite some relationships with CRP which is not a cytokine but rather an acute phase protein (lines 35 – 37). 

2.     Are three-day food records sufficient to determine DII especially in the setting of pregnancy where there are dynamic changes in the cytokine measures of interest in the second and third trimester? Repeated three-day record measures would help with interpretation – these seem to have been made but how often this was done is not clear from the methods.  

3.     At line 23 in the abstract and lines 392 & 393 in the conclusions section the authors claim that the changes in DII cause changes to cytokines but there is no evidence of this provided – it is an association.

4.     Page 4, section 2.5 – were the women fasted prior to sample collection?

5.     Given that age, education, household income, and pre-pregnancy second hand smoke exposure are correlated with DII, has the analysis of cytokines levels with DII been corrected for these likely confounders.  

6.     At lines 368 – 372 the authors make some suggestions regarding diet in pregnancy, but it is a stretch to base these recommendations on the findings of this study. 

Line 21 – ‘was’ should be ‘were’

Line 176 – ‘third and second’ should be ‘second and third’

Author Response

Thank you for taking the time to review our manuscript. Based on these comments and suggestions, we have made careful modifications to the original manuscript and carefully corrected the typographical errors.

Reviewer 2 Report

Very nice paper!

Author Response

Thank you for your careful reading and review of this manuscript; we have further polished the manuscript and made changes for better presentation.